# IIV-6 Inhibits NF-κB Responses in *Drosophila*

**DOI:** 10.3390/v11050409

**Published:** 2019-05-01

**Authors:** Cara West, Florentina Rus, Ying Chen, Anni Kleino, Monique Gangloff, Don B. Gammon, Neal Silverman

**Affiliations:** 1Division of Infectious Diseases and Immunology, Department of Medicine, University of Massachusetts Medical School, Worcester, MA 01605, USA; Cara.West@umassmed.edu (C.W.); Florentina.Rus@umassmed.edu (F.R.); anni.kleino@aias.au.dk (A.K.); 2RNA Therapeutics Institute, University of Massachusetts Medical School, Worcester, MA 01605, USA; cy4896@126.com; 3Department of Biochemistry, University of Cambridge, Cambridge CB2 1GA, UK; mg308@cam.ac.uk; 4Department of Microbiology, University of Texas Southwestern Medical Center, Dallas, TX T5390, USA; Don.Gammon@UTSouthwestern.edu

**Keywords:** viral immune evasion, immunomodulators, NF-κB, Imd, DNA virus, host-pathogen interactions, IIV-6

## Abstract

The host immune response and virus-encoded immune evasion proteins pose constant, mutual selective pressure on each other. Virally encoded immune evasion proteins also indicate which host pathways must be inhibited to allow for viral replication. Here, we show that IIV-6 is capable of inhibiting the two *Drosophila* NF-κB signaling pathways, Imd and Toll. Antimicrobial peptide (AMP) gene induction downstream of either pathway is suppressed when cells infected with IIV-6 are also stimulated with Toll or Imd ligands. We find that cleavage of both Imd and Relish, as well as Relish nuclear translocation, three key points in Imd signal transduction, occur in IIV-6 infected cells, indicating that the mechanism of viral inhibition is farther downstream, at the level of Relish promoter binding or transcriptional activation. Additionally, flies co-infected with both IIV-6 and the Gram-negative bacterium, *Erwinia carotovora carotovora*, succumb to infection more rapidly than flies singly infected with either the virus or the bacterium. These findings demonstrate how pre-existing infections can have a dramatic and negative effect on secondary infections, and establish a *Drosophila* model to study confection susceptibility.

## 1. Introduction

The host immune system and the viruses that challenge it face constant, mutual, selective pressure for survival. This perpetual arms race, known as the Red Queen Hypothesis [1], has created a plethora of mechanisms that the cell uses to thwart viral replication, and also a plethora of novel immune evasion tactics that viruses use to evade the immune response, in some cases even stealing genes from their hosts to suit this purpose [2]. In order to evade the innate immune response, viruses—especially large DNA viruses—encode a variety of immune evasion proteins to facilitate their replication. For example, the model poxvirus Vaccinia encodes a variety NF-κB pathway inhibitors, and a recently identified invertebrate DNA virus, Kallithea virus, is able to inhibit *Drosophila* Toll signaling [3,4]. Invertebrate Iridescent Virus-6 (IIV-6) is a large DNA virus capable of infecting Drosophila [5,6], with an estimated 215 open reading frames (ORFs) [7]. The large genome of IIV-6 suggests it encodes for many evasion proteins in addition to the characterized suppressor of RNAi (340L) [8], and Inhibitor of Apoptosis Protein (IAP) (193R) [9,10]. The interplay between the virus and host, particularly which pathways the virus devotes its resources to inhibit, provides insight as to which defense mechanisms pose the most threat to viral replication. A virus devoting its resources to shutting down a particular host pathway indicates that this pathway has applied selective pressure against the virus.

Whether the Toll and Imd pathways play a role in antiviral defense has been a topic probed in several studies, often with mixed or conflicting results [11,12,13,14]. Of course, these two NF-κB signaling pathways are well-known for their roles in anti-bacterial and anti-fungal defense, through the regulation of antimicrobial peptides (AMPs) and other effectors [15,16,17,18]. While some studies have argued these two pathways also have some antiviral activities, the antiviral mechanisms have not been elucidated [2,12,19,20,21]. To address the question of the potential anti-DNA-viral role of the Toll and Imd pathways, we took the approach of using the virus to illuminate which pathways might be antiviral by examining whether IIV-6 inhibits the Toll and Imd pathways. We found that cells infected with IIV-6 have suppressed AMP production, while other immune genes (i.e., *Turandots*, c-Jun N-terminal kinase (JNK) targets) remain induced, suggesting that this is not a global suppression of host transcription [22]. Surprisingly, the cleavage of both Imd and Relish, key signaling events in the Imd pathway [23,24], remain intact in IIV-6 infected cells. Relish nuclear translocation is also at least partly intact, indicating the blockage in Imd response likely occurs in the nucleus, at the level of promoter binding or transcriptional activation. The inhibition of NF-κB transcription factor activity may be more general, as the Toll pathway was also blocked by IIV-6 infection. Consistent with the inhibition of these critical antibacterial defense responses, flies infected with IIV-6 were hyper-susceptible to infection with the Gram-negative bacteria *Erwinia carotovora carotovora* (*Ecc15*), establishing a system for studying co-infection morbidity in the *Drosophila* model.

## 2. Materials and Methods

### 2.1. RNA Isolation and qRT-PCR

Total RNA from flies or S2* cells was extracted using TRIzol (Invitrogen, Waltham, MA, USA). Samples were then DNase treated (RQ1, Promega, Madison, WI, USA) and RNA re-extracted by phenol-chloroform. The cDNA was synthesized using iScript cDNA Synthesis kit (BioRad, Hercules CA, USA). Alternatively, the gDNAclear cDNA synthesis kit (BioRad) was used following TRIzol purification. The quantitative reverse transcriptase–polymerase chain reaction (qRT-PCR) was analyzed by normalizing to the housekeeping gene *Rp49*.

### 2.2. The nCounter Analysis

The expression levels of 102 *Drosophila* immune genes (Appendix A) were assayed from 100 nanograms of RNA via a customized Nanostring nCounter codeset. Two biological replicates of S2* cells were analyzed for each treatment and timepoint. The results were analyzed using nSolver 4.0 software according to the manufacturer’s instructions (NanoString Technologies, Seattle, WA, USA), and the heatmap was created using nSolver 4.0 software and JavaTree. Shown are the AMPs, p38, JNK, and JAK-STAT targets with counts above background. Figure 3 analyzes data previously presented in West et al. [22].

### 2.3. Fly Stocks and In Vivo Studies

Three to five day old *w*^1118^ flies, of approximately equal proportions male and female, maintained at 22 °C, were used for all experiments. Flies were injected intrathoracically with 32.2 nL of virus (1 × 10^4^ TCID50) or vehicle (PBS) using a Nanoject II (Drummond, Broomall, PA, USA). For survival assays, a minimum of fifty flies were used per treatment, per genotype, and the dead were counted daily. Kaplan-Meier curves are shown and significance was determined by Mantel-Cox log-rank using GraphPad Prism.

*Erwinia carotovora carotovora 15*, also known as *Pectobacterium carotovora*, was cultured overnight by shaking at 250 RPMs in LB broth, pelleted at 13,000× *g* for 10 min in a microfuge, washed in PBS, and re-pelleted. Infections were performed by dipping a microsurgery needle into the concentrated bacterial pellet, and pricking in the thorax. The IIV-6 injection site was identified using melanization, and bacterial pricking was performed on the opposite side of the thorax.

### 2.4. Cell Culture

S2* cells were cultured as previously described [25,26] and were treated with 1 μM 20-hydroxyecdysone, (EcD) for 18 h followed by infection with IIV-6 at a multiplicity of infection (MOI) of 2 for 6 h. Cells were then stimulated with 2 μg/mL PGN for 6 h to stimulate the Imd pathway, or stimulated with cleaved Spätzle for 18 h for Toll pathway stimulations, and were harvested for RNA isolation. 

### 2.5. Immunoblots

S2* cells were cultured and treated as described above. For protein analysis, cells were then stimulated with 2 μg/mL PGN for 15 min to stimulate the Imd pathway, and harvested in lysis buffer consisting of 1% Triton X-100, 20 mM Tris pH 7.5, 150 mM NaCl, 2 mM EDTA, and 10% glycerol, with 1 mM DTT, 1 mM Sodium Vandate, 20 mM β-glycerolphosphate, and protease inhibitors. Imd or c-Rel antibodies were used as previously described [24,27].

### 2.6. Confocal Microscopy

YFP-Relish over-expressing cells were kept in selection media with hygromycin (200 μg/mL), and cultured as previously described [28]. Cells were treated with EcD for 2 h before being infected with mCherry-IIV-6 at an MOI of 2 for 18 h. Cells were then plated on Alcian Blue or ConA treated coverslips for 30 min and stimulated with PGN for 15 min. Cells were fixed in 4% paraformaldehyde, and stained with anti-lamin Dm0 (Developmental Studies Hybridoma Bank, ADL84.12) and Hoechst 33342.

### 2.7. Virus Preparation

IIV-6 was provided by Luis Teixeira. IIV-6 was propagated and purified on DL-1 cells as previously described [5], with a final resuspension in PBS, and quantified on DL-1 cells by TCID50. Cells were infected at an MOI of 2 unless otherwise noted, while flies were injected with 1 × 10^4^ TCID50, as detailed above.

The ΔTS-MCP-mCherry-IIV-6 (mCherry-IIV-6) was created by inserting mCherry under the control of the major capsid promoter into the thymidylate synthase locus, using homologous recombination [29]. The major capsid promoter sequence was designed based on the analysis of IIV-6 MCP regulatory sequences by Nalçacioǧlu et al. [30] 

## 3. Results

### 3.1. AMP Production is Suppressed in the Presence of IIV-6

Given the scarce and somewhat conflicting data on the antiviral effects of the *Drosophila* NF-κB pathways in response to viral infections [2,12,19,20,21], we tested whether IIV-6 inhibits the Imd or Toll pathways. S2* cells were differentiated with 1 μM 20-hydroxyecdysone (EcD) for 18 h followed by IIV-6 infection for 6 h. Treatment with EcD is required to differentiate these cells and induce the expression of the receptor PGRP-LC, making these cells highly responsive to DAP-type peptidoglycan (PGN) stimulation [28]. After 6 h of IIV-6 infection, cells were then stimulated with 2 μg/mL of PGN for an additional 6 h, and gene expression analyzed by qRT-PCR. IIV-6 infection reduced transcriptional induction of the Imd-dependent AMP gene *Diptericin* to near background levels (Figure 1A).

To determine if IIV-6 more generally blocks NF-κB signaling, we similarly analyzed Toll signaling. S2* cells were cultured in a similar manner, treated with EcD for 18 h, which upregulates expression of the Toll receptor [28], infected with IIV-6 for 6 h, and then stimulated with the Toll ligand, cleaved Spätzle, for 18 h. Induction of the Toll target gene *Drosomycin* was strongly inhibited by IIV-6 infection, as analyzed by qRT-PCR (Figure 1B). To examine both of these NF-κB pathways more broadly, we utilized NanoString nCounter Analysis, with a custom designed codeset probing over 100 immune-related genes, including most known AMP genes. All of the antimicrobial peptides were down-regulated in IIV-6 infected samples when the Imd pathway was stimulated with PGN (Figure 2A, Appendix A) or Toll pathway stimulated with cleaved Späztzle (Figure 2B, Appendix A), compared to the mock-infected controls stimulated with Imd or Toll ligands. Notably, other groups of genes were induced by virus infection, including the JNK targets *puckered* and *punch*, and the p38-dependent genes *upd3* and *Ddc* [22,31]. The antiviral gene *Ars2* was also induced by IIV-6 [32]. Since JNK signaling can be initiated through TAK1, the fact that JNK targets are being transcribed suggests that Imd signaling is being initiated, and signaling successfully occurs through at least TAK1 [33]. The induction of these genes by IIV-6 argues that the virus is not simply shutting down all host transcription, but is specifically targeting the two NF-κB signaling pathways, Imd and Toll. Additionally, we saw no suppression of *GAPDH1* levels (Appendix A).

IIV-6 inhibition of both NF-κB pathways suggests that these pathways may function to limit viral replication. To examine this possibility, we analyzed an earlier Nanostring dataset from adult flies infected with IIV-6, or injected with a similar volume of PBS, to determine if virus infection induces any of the AMP genes [22]. Indeed, IIV-6 modestly induced AMP gene expression two-fold over PBS-injected controls 12 hours post-infection (Figure 3A,A’). However, by 24 h, this induction has returned to, or in some instances below, baseline levels. These findings are consistent with an early AMP response, possibly through NF-κB signaling pathways, which is quickly extinguished by viral inhibitors.

### 3.2. NF-κB Inhibition is Downstream of Imd and Relish

Next, we focused on the Imd pathway to tease apart the mechanism of NF-κB inhibition by IIV-6. As Imd signaling requires cleavage of Imd by the caspase 8-like Dredd [24,34], we probed for cleaved Imd in IIV-6 infected or uninfected cell lysates. We found robust cleavage of Imd upon PGN stimulation in both the presence and absence of IIV-6, indicating that the blockage of Imd signaling is likely to occur downstream of Imd cleavage (Figure 4A). To probe further downstream, we examined Relish cleavage, another critical event in Imd signaling, utilizing a C-terminal Relish antibody. As expected, samples mock treated or infected with IIV-6 alone show a prominent 110 kD band indicating full-length Relish, and control samples treated with EcD and PGN show complete processing of full-length Relish. Likewise, upon PGN stimulation, Relish was fully cleaved in the presence of IIV-6, even at a relatively high MOI (Figure 4B, lanes 4–6). These results indicate the inhibition of Imd signaling by IIV-6 occurs downstream of Relish cleavage, the key event in the activation of this NF-κB precursor.

Since Relish is cleaved in the presence of IIV-6, we next sought to determine whether IIV-6 inhibits Imd signaling through blocking Relish nuclear translocation. To this end, we utilized a YFP-Relish over-expressing cell line, and a strain of IIV-6 expressing mCherry under the control of the major capsid protein (MCP). Even with these tools, scoring Relish nuclear translocation after virus infection presented several technical challenges. In particular, viral infected cells adhered poorly to treated coverslips and displayed reduced YFP-Relish signal. However, the levels of endogenous Relish protein are not altered by IIV-6 infection (Figure 4B). Additionally, IIV-6 undergoes massive DNA viral replication in the cytoplasmic viral factories, making it difficult to distinguish the nucleus. In order to identify the nucleus, cells were stained with both Hoechst 33342 and the nuclear envelope marker Lamin to accurately discern the nucleus from the cytosolic viral factory. In control conditions, cells stimulated with PGN showed 35% nuclear translocation, while Relish remained cytoplasmic in 20% of cells (Figure 5A, lower left panel, Figure 5B). The remaining 44% of cells contained a YFP signal in both the cytoplasm and nucleus or had a YFP signal too weak to determine localization. In IIV-6 infected cells, we found that approximately 17% showed nuclear translocation, with 33% remaining cytoplasmic, while nearly 50% displayed both cytoplasmic and nuclear Relish or had a YFP signal too weak to determine localization (Figure 5A lower right panel, Figure 5B). While IIV-6 reduces the detected level of PGN-induced Relish nuclear translocation compared to uninfected PGN-stimulated cells, this difference was not significant, suggesting PGN can still trigger some Relish nuclear translocation in virus infected cells. Interestingly, virus infection alone caused some Relish translocation (Figure 5A upper right panel, Figure 5B). These data suggest that IIV-6 does not completely inhibit Relish nuclear translocation, and that viral suppression of Imd signaling likely occurs downstream, either at the level of Relish promoter binding or transcriptional activation.

### 3.3. Inhibition of NF-κB Signaling is Mediated by an Immediate Early Gene

Most—but not all—immune evasion proteins are immediate early genes. Strategically, it is probably most effective, from the virus perspective, to shut down the host defense as quickly as possible and rapidly achieve high levels of replication. In order to determine if IIV6 inhibition of *Drosophila* NF-κB signaling also involves an early gene product, cells were treated with the viral polymerase inhibitor cidofovir, or infected with heat- or UV-inactivated virus, and then stimulated with PGN to probe Imd signaling. PGN-triggered *Diptericin* induction was blocked with all three virus-inactivating treatments (Figure 6), indicating that the IIV6-mediated NF-κB inhibition was the result of immediate early genes, or possibly associated with a factor directly delivered with the virion.

### 3.4. Flies Infected with IIV-6 are more Susceptible to Bacterial Infection

Together, these results indicate that infection with IIV-6 results in the global suppression of NF-κB signaling in flies—a major component of the innate immune response to bacterial and fungal infections in flies and mammals. Consistent with our in vitro data showing universally suppressed AMPs, we found that flies infected with IIV-6 also had suppression of NF-κB signaling, as indicated by lower mRNA levels of the AMP genes *Diptericin* and *Drosomycin* compared to PBS-injected or unmanipulated flies at day 8 post-infection (Figure 7A, Appendix A). This suggests that flies infected with IIV-6 should be highly susceptible to other microbial infections. To test this hypothesis, we infected *Drosophila* adults with IIV-6 for seven days. On day eight post-IIV-6 infection, we pricked one group with a sterile needle, and the other group of flies with a needle dipped in *Erwinia carotovora carotovora* (*Ecc15*), a Gram-negative pathogen with limited lethality in healthy, immunocompetent flies. Seven days after this secondary bacterial infection, 50% of the flies infected with both IIV-6 and *Ecc15* had succumbed to infection, with nearly 100% lethality by day 20 (Figure 7B). In contrast, flies that had been mock-injected with PBS prior to *Ecc15* infection had a median survival time of twenty-three days post-secondary (bacterial) infection. The animals injected with IIV-6 followed by a clean prick reached 50% lethality only at day 18, typical of the IIV-6 survival curve [22]. Together, these data show that an underlying IIV-6 infection can have a dramatic effect on a secondary bacterial infection, causing flies to be far more susceptible to *Ecc15* infection than unaffected animals. Given the critical role of the Imd pathway in defending against *Ecc15* infection, these results are consistent with the inhibition of Imd signaling observed in IIV-6 infected S2* cells and adult flies.

## 4. Discussion

Here, we show that IIV-6 infection interferes with the antimicrobial peptide response mediated through the Toll and Imd NF-κB pathways. In addition, we find that flies infected with IIV-6 succumb to an otherwise mildly pathogenic bacterial infection in vivo. Together, our data show that an underlying infection with this DNA virus can dampen the immune response and dramatically alter the outcome of a secondary bacterial infection, turning an otherwise innocuous infection into a lethal one.

While it appears that IIV-6 is suppressing the AMP response by actively inhibiting the NF-κB pathways, the mechanisms of inhibition remain unclear. It is possible that IIV-6 encodes an NF-κB inhibitor that prevents Relish binding to κB sites or recruitment of polymerase machinery. Several examples of viral-encoded NF-κB inhibitory proteins acting within the nucleus have been reported. NF-κB inhibitors functioning within the nucleus are encoded by vaccinia virus (VACV) [35], as well as African swine fever virus A238L [36]. VACV also encodes an inhibitor of Type 1 IFN, C6, that functions post-STAT1 and 2 nuclear translocation and DNA binding by binding the STAT2 transactivation domain [37]. Another VACV protein, N2 inhibits IRF3 within the nucleus [38]. Future studies will be aimed at examining the effect of IIV-6 on Relish binding at AMP gene κB sites, as well as the transcriptional activation process. 

The timing of suppression also needs to be examined more closely. While our in vivo or in vitro data may suggest a difference in the suppressive effects of IIV-6, it is important to note that a more detailed in vivo time course is required. If there is a true shift in suppression, we hypothesize that this may be due to the fact that the in vitro infections occur at an MOI of 2, and we can more easily synchronize the timing of infection. In vivo, it may be that virus takes longer to diffuse through the hemolymph and replicate, resulting in a difference in response.

It should be noted that IIV-6 does not encode a homolog of Diedel, the putative Imd inhibitor, as other large insect DNA viruses do [2]. Given the role of RHIM-dependent amyloid fibrils in the Imd pathway [39], we also considered that IIV-6 might encode RHIM-inhibitor, similar to Murine cytomegalovirus M45 [40], but found no evidence for such a factor in the IIV-6 genome.

Future studies will be required to determine which viral gene(s) are responsible for NF-κB inhibition. Our work shows that Imd signaling is inhibited at a step downstream of Relish activation. However, the mechanisms of Toll inhibition have not been investigated yet. IIV-6 may encode more than one NF-κB inhibitor, targeting the Imd and Toll pathways at different points. Since other large DNA viruses, such as VACV, encode multiple inhibitors targeting NF-κB pathways at various points in mammalian NF-κB signaling pathways [3], elimination of single viral genes may not reveal any phenotype due to redundancies. Determining which viral genes are responsible for NF-κB inhibition will require both loss and gain of function approaches and will be the subject of future studies. 

IIV-6 has a broad host range, infecting a large variety of insects with agricultural and economic importance. The slow replication cycle of this virus results in infected insects surviving for weeks, allowing it to spread amongst a population, and consequently, leaving that population with an increased susceptibility to a secondary bacterial infection that would otherwise be cleared. Given the current decline of honey bees, as well as the entire insect population [41], a persistent viral infection could further damage this already precarious population by increasing susceptibility to a range of other microbial pathogens. While previous studies have ruled out IIV-6 as the causative agent of colony collapse disorder [42], it should be noted that IIV-6 infection may expose other species to increased vulnerability to a secondary infection. 

In summary, we have shown that IIV-6 infection results in an inability to mount an NF-κB mediated AMP response upon stimulation of either the Imd or Toll pathways. It appears that IIV-6 infection may briefly induce AMP expression before shutting it down. Suppression of Imd signaling occurs downstream of Relish activation, suggesting a viral inhibitor may act at the level of DNA binding and transcriptional activation. Nanostring data showing that IIV-6 induces transcription of JNK targets suggest that the Imd pathway is being successfully activated through at least the branch point of TAK1 [25]. These results are consistent with the data showing that Relish cleavage and some nuclear translocation remain intact and argue that the virally generated block occurs far downstream in this pathway. Strikingly, flies infected with IIV-6 subjected to subsequent infection with *Ecc15* died more rapidly than flies singly infected with virus or bacteria, indicating that this virus-mediated block in NF-κB signaling may have a profound effect on resistance to bacterial infections. 

The involvement of the Drosophila NF-κB pathways in antiviral signaling is a controversial topic. Other studies have suggested an antiviral role for AMPs, examining a recombinant analog of the RNA virus Sindbis virus [21]. Whether or not some AMPs are also potent against DNA viruses, such as IIV-6, will be the focus of future studies, for example by examining the replication of mCherry -expressing virus or by quantifying viral loads from infected flies over-expressing AMPs. It is also possible that AMP induction is simply a by-product of Relish activation, similar to many ISGs that have no direct protective role against the pathogen eliciting their expression. In this case, other Relish target genes may be important for antiviral defense. Additionally, with some picrorna-like viruses, such as Drosophila C virus and Cricket Paralysis virus, some—but not all—Imd pathway components seem to be restrictive [12,13]. Identifying a viral-encoded Imd inhibitor would provide more evidence supporting that this pathway induces antiviral activities. However, how Imd signaling is initiated upon virus infection, when the established agonist for this pathway is bacterial peptidoglycan [43,44], and whether all Imd signaling components are necessary, need to be examined in greater detail. For example, as the key receptors in this pathway, PGRP-LC and PGRP-LE, directly bind to DAP-type PGN, it seems unlikely that they will be involved in viral recognition. Antiviral studies implicating Imd signaling should be reevaluated in light of recent work showing that the microbiome plays an important role activating Imd signaling in the gut to prime an antiviral response via ERK following an oral infection route [45]. Whether commensals can act to influence antiviral responses in other organ systems requires further exploration. For example, the Malpighian tubules are organs branching off from the *Drosophila* gut, which absorb waste from the hemolymph, but have also been shown to serve important roles in immune function, including induction and secretion of AMPs in response to infection [43,46,47,48]. Whether the Malpighian tubules provide crosstalk between the gut and hemolymph during viral infections should be explored.

## Figures and Tables

**Figure 1 viruses-11-00409-f001:**
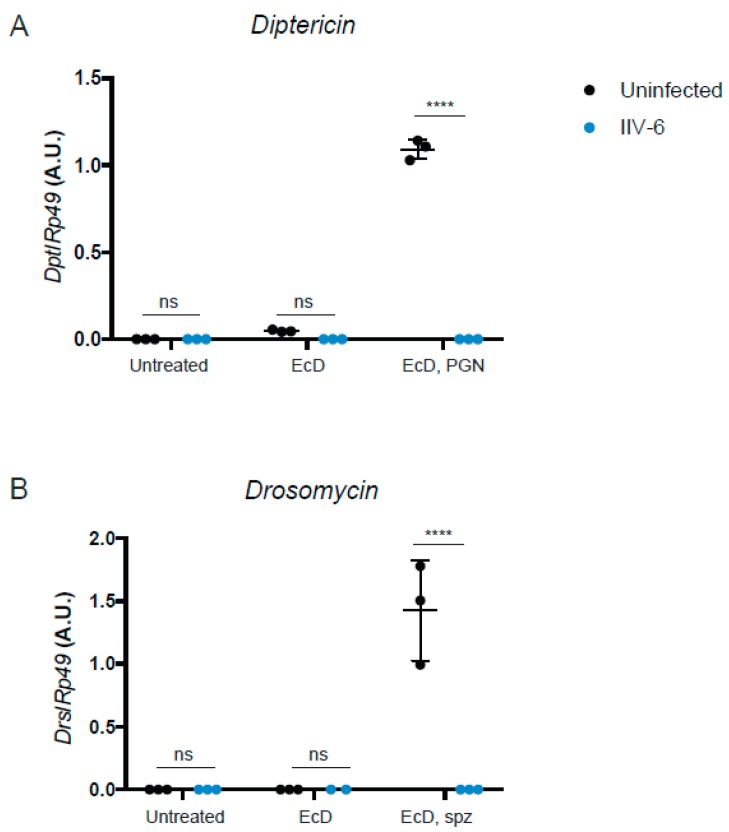
IIV-6 inhibits Imd and Toll Signaling. (**A**) S2* cells were treated with 20-hydroxyecdysone (EcD) as indicated for 18 hours and then infected with IIV-6 (blue circles) or uninfected (black circles) for six hours. Cells were then stimulated with DAP-type PGN for six hours, where indicated. *Diptericin* (*Dpt*) levels were monitored by qRT-PCR. (**B**) S2* cells were treated with 20-hydroxyecdysone (EcD) as indicated for 18 h and then infected with IIV-6 (blue circles) or uninfected (black circles) for six hours. Cells were then stimulated with cleaved Spätzle (spz) for 18 h, where indicated. *Drosomycin* (*Drs*) levels were monitored by qRT-PCR. (**A**,**B**) Black bars indicate mean and error bars indicate standard deviation. Statistics were determined using two-way ANOVA and Sidak’s multiple comparisons test; ns, not significant; ****, *p* < 0.0001.

**Figure 2 viruses-11-00409-f002:**
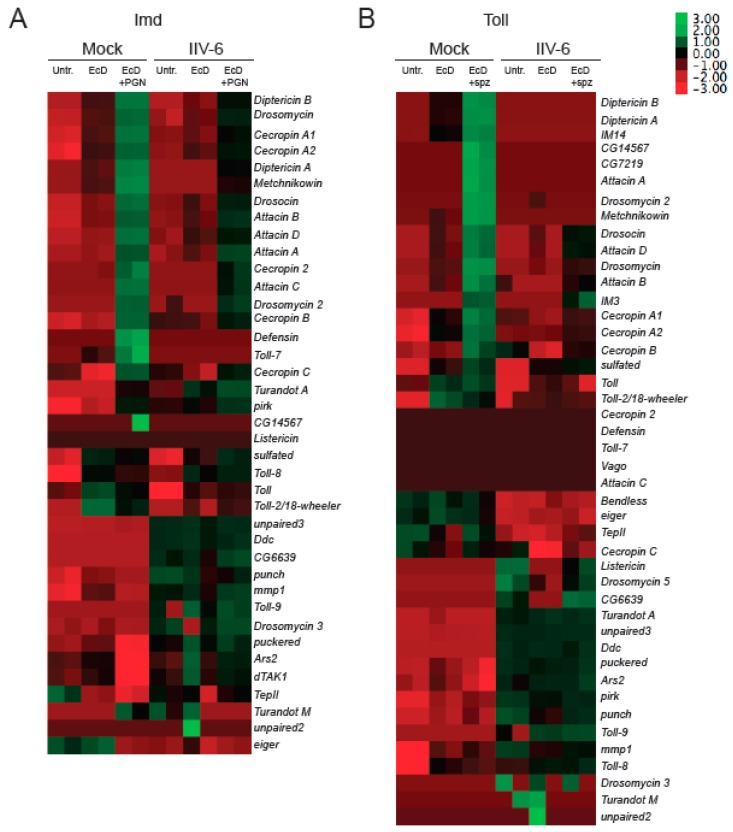
Both Imd and Toll regulated AMPs are suppressed by IIV-6 infection. S2* cells were treated with 20-hydroxyecdysone (EcD) for 18 h, and then infected with IIV-6 for 6 h. (**A**) Cells were stimulated with peptidoglycan (PGN) for 6 h prior to RNA isolation. (**B**) S2* cells were then stimulated with cleaved Spätzle (spz) for 18 h prior to RNA isolation, and then analyzed by Nanostring nCounter. Heatmaps display Z scores of mRNA levels of immune genes in the presence or absence of virus and pathway stimulation clustered by expression pattern. Biologically independent duplicates are shown. Untr., untreated cells. EcD, cells treated with 20-hydroxyecdysone. EcD + PGN, cells treated with 20-hydroxyecdysone and peptidoglycan. EcD + spz, cells treated with 20-hydroxyecdysone and Spätzle.

**Figure 3 viruses-11-00409-f003:**
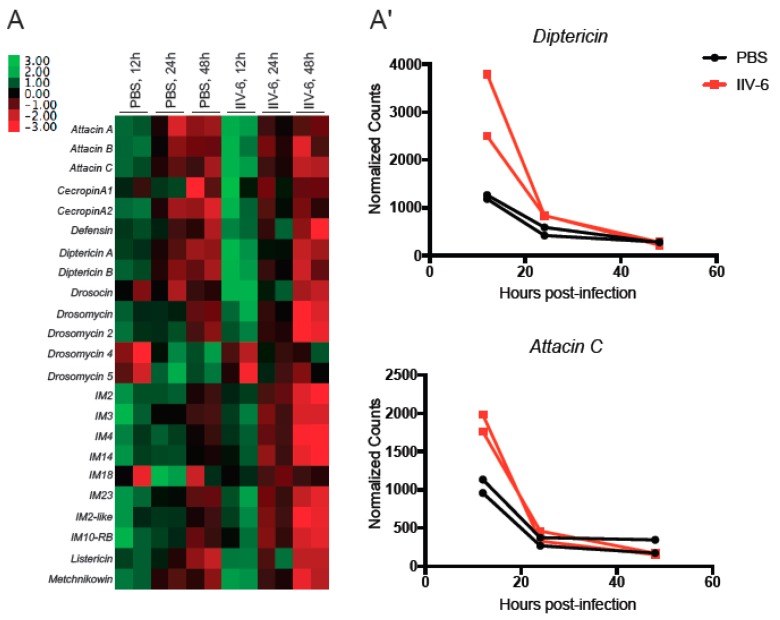
Some AMPs are elevated in vivo upon IIV-6 infection, before returning to baseline. (**A**) Heatmap of Z score transformed mRNA levels for AMP genes following IIV-6 infection of adult male *w*^1118^ flies for the indicated timepoints, assayed by Nanostring nCounter. RNA was isolated from PBS-injected flies at the same time points as a control. Biologically independent samples were analyzed in duplicate. (**A’**) Detailed comparison of mRNA levels for selected Imd-regulated AMP genes following IIV-6 infection of adult *w*^1118^ flies for 12, 24, and 48 h assayed by Nanostring nCounter.

**Figure 4 viruses-11-00409-f004:**
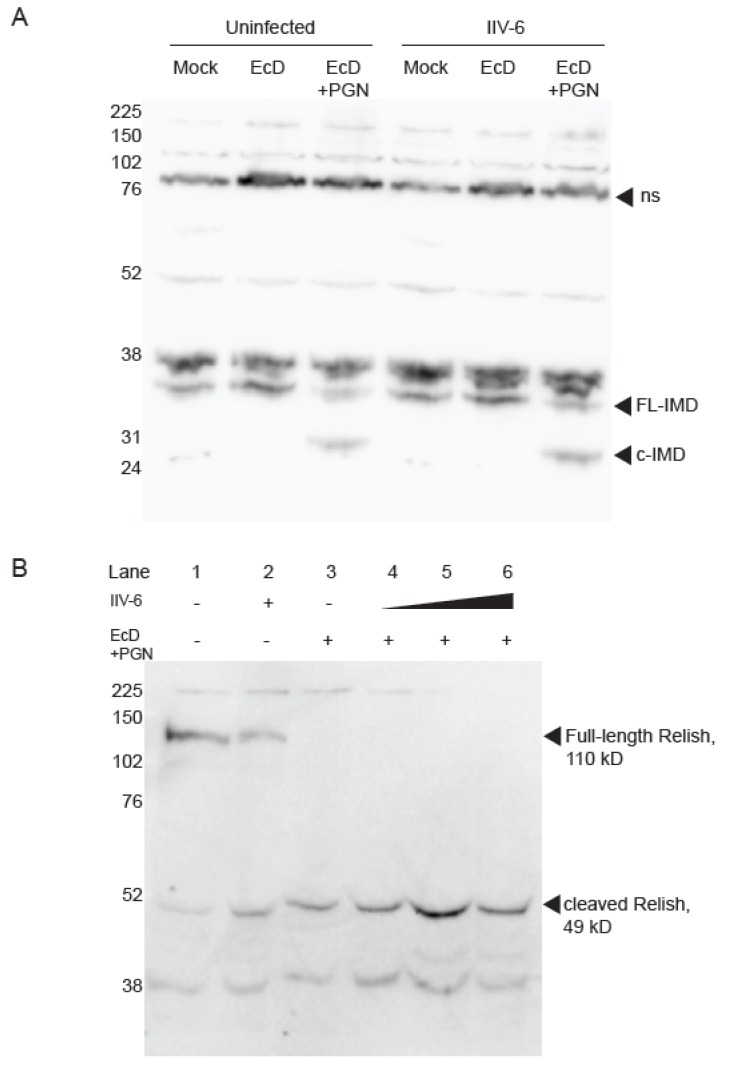
Imd signaling remains intact in the presence of IIV-6. (**A**) S2* cells were treated with 20-hydroxyecdysone (EcD) for 18 h. Cells were then infected with IIV-6, where indicated, for six hours. Samples were then stimulated with PGN for 15 min, where indicated, and lysed in standard lysis buffer. Endogenous Imd was monitored by immunoblotting; arrows indicate cleaved (c-Imd) or full-length (FL-Imd) Imd; ns, non-specific. (**B**) Cells were treated with 20-hydroxyecdysone (EcD), where indicated, for 18 h and infected with IIV-6, as indicated, for six hours. Samples stimulated with EcD were then stimulated with PGN for 15 min, and lysed in standard lysis buffer. Endogenous Relish was probed by immunoblotting using a C-terminal Relish antibody. MOI used was as follows. Lane 2: MOI =2; Lane 4: MOI = 0.2; Lane 5: MOI = 2; Lane 6: MOI = 5.

**Figure 5 viruses-11-00409-f005:**
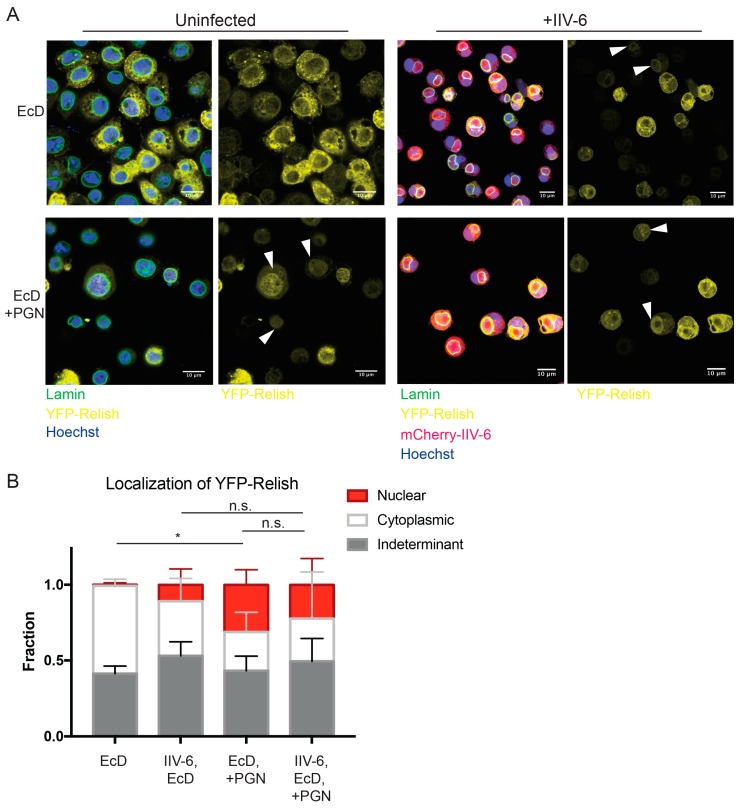
Nuclear localization of Relish remains intact in the presence of IIV-6. (**A**) S2* cells stably expressing YFP-Relish were treated with 20-hydroxyecdysone (EcD) for 2 h and then infected with mCherry-IIV-6 (right) or left uninfected (left) for 18 h. Cells were then stimulated with DAP-type PGN for 15 min (lower panels), prior to fixation and staining with anti-Lamin (shown in green) and Hoechst 33342 (shown in blue). Representative images from four biologically independent experiments are shown. Arrowheads mark cells with nuclear localized Relish. Left panels display the overlay of YFP-Relish, Hoeschst, Lamin, and mCherry-IIV-6 where applicable, while the right panels exhibit the YFP-Relish alone. Scale bars are 10 μm. (**B**) Quantification of Relish nuclear translocation. Between 400–1800 cells were scored for each condition as displaying Relish localization in the nucleus, in the cytoplasm, or as indeterminate when staining was diffuse throughout both compartments or the signal was too weak to discern. Statistics were determined using two-way ANOVA and Tukey’s multiple comparisons test; *, *p* < 0.05; ns, not significant. Error bars represent standard deviation.

**Figure 6 viruses-11-00409-f006:**
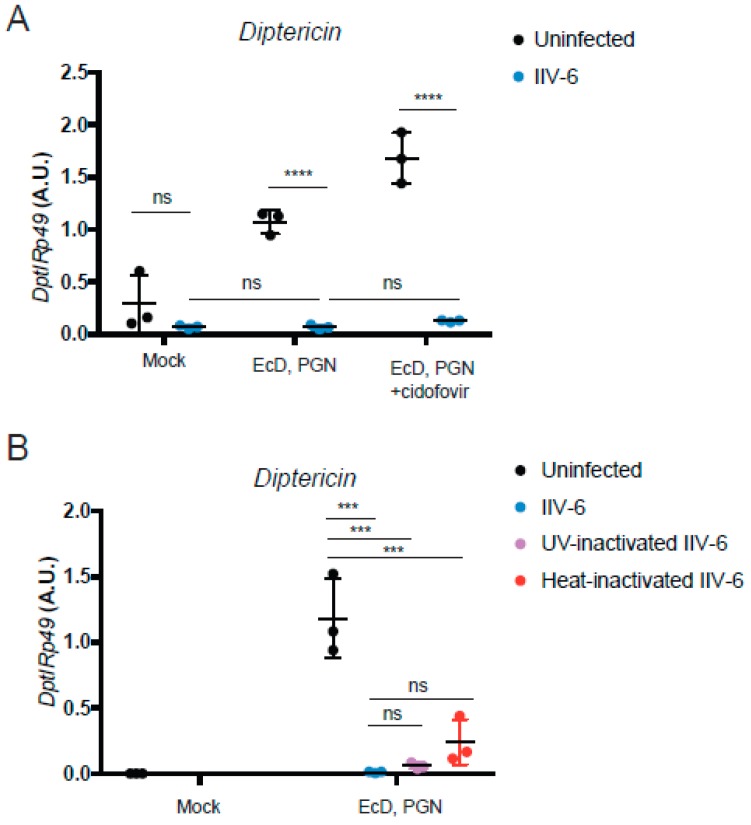
Viral replication is not needed for NF-κB inhibition. (**A**,**B**) S2* cells were treated with 20-hydroxyecdysone (EcD) as indicated for 18 h and then infected with IIV-6 (blue circles) or uninfected (black circles) for six hours. Cells were then stimulated with DAP-type PGN for six hours, where indicated. *Diptericin* levels were monitored by qRT-PCR. (**A**) Cells were treated with cidofovir, a viral polymerase inhibitor, where indicated. Mock cells untreated with EcD, PGN, or cidofovir are shown as a control. Each data point is a biologically independent replicate; *n* = 3. Error bars represent standard deviation. Statistics were determined using two-way ANOVA and Sidak’s multiple comparisons test; ns, not significant; ****, *p* < 0.0001. (**B**) S2* cells were infected with IIV-6 (blue circles), or treated with UV- (purple circles) or heat- (red circles) inactivated IIV-6. Uninfected controls are shown in black. Each data point is a biologically independent replicate; *n* = 3. Error bars represent standard deviation. Statistics were determined using one-way ANOVA and Tukey’s multiple comparisons test; ns, not significant; ***, *p* < 0.001.

**Figure 7 viruses-11-00409-f007:**
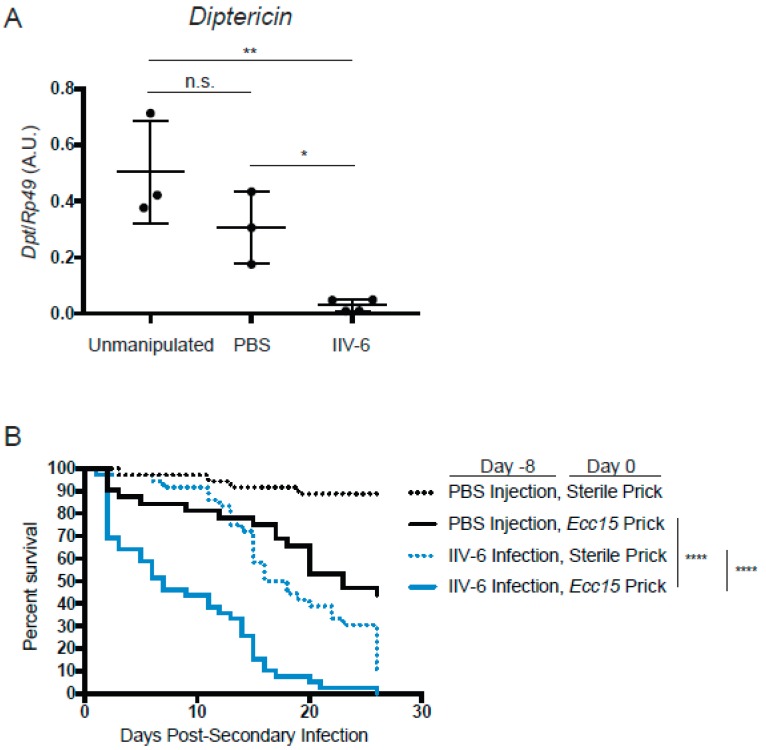
Flies infected with IIV-6 have lower AMP levels. (**A**) Adult *w*^1118^ flies were infected with IIV-6, injected with PBS, or left unmanipulated for 7 days, and then snap frozen for RNA isolation. *Dpt* levels were assayed by qRT-PCR. Black bars indicate mean and error bars indicate standard deviation. Statistics were determined using one-way ANOVA and Tukey’s multiple comparisons test; ns, not significant; *, *p* = 0.0463; **, *p* = 0.0033. (**B**) Kaplan-Meier plots of adult *w*^1118^ flies infected with IIV-6 (blue lines) or PBS-injected (black lines) for 8 days prior to bacterial infection. On day 0, flies were pricked with *Ecc15* (solid lines) or were sterile pricked (dashed lines) with a microsurgery needle. Flies were counted daily for survivors. Statistics were determined using log-rank test; ****, *p* < 0.0001.

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
