# Peer review of "IIV-6 Inhibits NF-κB Responses in Drosophila"

_viruses, 2019, doi:10.3390/v11050409_

Round 1

Reviewer 1 Report

In this study, West et.al. found that IIV-6 could inhibit the Drosophila NF-κB signaling pathway and flies infected with IIV-6 are more susceptible to bacterial infection. The strength of the study is to touch the hot topic of innate immunity. Generally, the figures are easy to follow and data are well organized. However, there are some concerns needing to be addressed.
1. In Figure 1, for excluding the possibility that IIV-6 infection suppresses the transcription of all genes, it's better to detect another gene, which are not inhibited by IIV-6 infection. 

2. In Figure 7A, the authors need to detect the expression of another gene Drosomycin.

Author Response

In this study, West et.al. found that IIV-6 could inhibit the Drosophila NF-κB signaling pathway and flies infected with IIV-6 are more susceptible to bacterial infection. The strength of the study is to touch the hot topic of innate immunity. Generally, the figures are easy to follow and data are well organized. However, there are some concerns needing to be addressed.

In Figure 1, for excluding the possibility that IIV-6 infection suppresses the transcription of all genes, it's better to detect another gene, which are not inhibited by IIV-6 infection. 

We have shown that other genes are not inhibited by IIV-6 induction in our Nanostring data in Figure 2. Additionally, all gene expression levels measured by qRT-PCR have been normalized to the housekeeping gene Rp49, thus correcting for any overall changes to host gene transcription. In addition, we now included analysis GAPDH levels (also normalized to Rp49) as Supplemental Figure 1.

2. In Figure 7A, the authors need to detect the expression of another gene Drosomycin.

We have now measured Drosomycin levels in these samples and nowinclude them as Supplemental Figure 2.

Reviewer 2 Report

I have read the manuscript with interest. The main finding of this manuscript is that the DNA virus IIV-6 is capable of inhibiting NF-kB signaling pathways in Drosophila. AMP gene expression upon infection with IIV-6 is suppressed in the presence of Toll and IMD ligands. The present study uses both in vitro and in vivo assays to characterize the site of viral inhibition in Drosophila and places it at the level of Relish promoter binding or transcriptional activation. The authors show elegantly that no viral replication is required for the suppression, suggesting the involvement of immediate early genes of IIV-6 in the suppression effect. Additionally, flies co-infected with IIV-6 and the entomopathogenic bacterium ECC succumb faster than monoinfected flies suggesting some potentials to develop a Drosophila model to study the immune response against multiinfections.

General comments:

It is striking to see the effect of IIV-6 on AMP expression, although targets of AMPs are thought to be mainly bacteria and fungi. It is tempting to speculate if AMPs in return have a direct effect on IIV-6. What I would have liked to see is to test overexpression of an AMP, possibly diptericin or drosomycin, upon IIV-6 infection. Would an overexpression of one of those AMPs have a negative effect on IIV-6, like decreasing the viral load per fly? I could imagine that the generation of da-GAl4 > UAS-Dipt or da-GAl4 > UAS-Drs flies, the infection of those flies with IIV-6 and performing RT-qPCR to measure the viral load could resolve this question. If the experiment should not for any reason be feasible, a thought experiment dealing with this point in the discussion section would be highly appreciated.

Compared to the in vitro data, the analysis using in vivo model of a previous nanostring data set (Fig. 3) suggest a shift in the timing of suppression. This issue should be shortly described in the discussion section.

I have some difficulties to discern the outcome of the nuclear localization of Relish. As the authors mentioned, the analysis faced several technical challenges. It seems that the number of IIV-6 infected cells is generally lower than in the uninfected state. The infection might change the properties of the cells, rendering them more sensitive to the preparation procedure. Probably, the YFP-Relish signal is clearer in the original images, but in the online and printed version, I have really difficulties to see the localization of YFP, except in the uninfected state where it is located in the cytoplasm. Thus, I would suggest adding panels just displaying the YFP, maybe as an inset in the existing images.

It is generally well written, although one text part in the material and methods section seem to be copy-pasted and should therefore be changed for stylistic reason (L320-L322 and L326-L328).

 Specific comments:

1.       line 27: “…perpetual arms race has created a plethora of mechanisms…” - I would suggest to introduce the term Red Queen Hypothesis from Van Valen (1973, A new evolutionary law. Evolutionary Theory 1: 1–30. ) in this context.

2.       line 34: the information that IIV-6 can infect Drosophila melanogaster should be added, also the reference (Bronkhorst et al., PNAS, 2012: The DNA virus Invertebrate iridescent virus 6 is a target of the Drosophila RNAi machinery).

3.       line 46: Please remove “In order to begin” and start the sentence directly with “To address…”. The whole sentence is a bit bulky; please consider rephrasing it.

4.       line 51:  I am not familiar with the term cellular catastrophe. Could you please explain this term or use another word?

5.       line 81 – line 82: Is the treatment with EcD also required for the expression of the Toll receptor? If this is the case, please add a short comment. If not, please explain why you add the treatment.

6.       line 85: The codeset of 139 immune-related genes should be included as list in the supplementary materials.

7.       line 102: Please add in Fig. 2 a short description of the color code (-3.00 to 3.00 as level of expression). Do this also for Fig. 3.

8.       line 104: Add the abbreviation for EcD+PGN and EcD+spz.

9.       line 106: Remove “To begin”.

10.   line 108: Full stop/ period is missing after AMP genes [19].

11.   line 125: Replace “must” with “is likely to”.

12.   line 136: Fig. 4, why do you change the experimental design and stimulate the samples just for 15 minutes? In the experiments before, the stimulation was 6 hours. Please explain.

13.   line 148: Please remove the brackets and rephrase the sentence in the bracket to fit in the text.

14.   line 162: Replace “argue” with “suggest”.

15.   line 214: Full stop/ period is missing after “survival curve [19]”.

16.   line 309: The outcome of infection survivals depend also on the gender utilized. Please specify the gender you have used, and if you used males and females in the same survival, the ratio between the gender.

17.   line 314 - 315: At which incubation temperature was ECC15 grown? Did you use a shaker, and if so, which rpm did you use? At which speed did you spun the culture?

18.   line 329: Please specify “standard lysis buffer”.

19.   line 337: which Hoechst did you use?

Author Response

I have read the manuscript with interest. The main finding of this manuscript is that the DNA virus IIV-6 is capable of inhibiting NF-kB signaling pathways in Drosophila. AMP gene expression upon infection with IIV-6 is suppressed in the presence of Toll and IMD ligands. The present study uses both in vitro and in vivo assays to characterize the site of viral inhibition in Drosophila and places it at the level of Relish promoter binding or transcriptional activation. The authors show elegantly that no viral replication is required for the suppression, suggesting the involvement of immediate early genes of IIV-6 in the suppression effect. Additionally, flies co-infected with IIV-6 and the entomopathogenic bacterium ECC succumb faster than monoinfected flies suggesting some potentials to develop a Drosophila model to study the immune response against multiinfections.

General comments:

It is striking to see the effect of IIV-6 on AMP expression, although targets of AMPs are thought to be mainly bacteria and fungi. It is tempting to speculate if AMPs in return have a direct effect on IIV-6. What I would have liked to see is to test overexpression of an AMP, possibly diptericin or drosomycin, upon IIV-6 infection. Would an overexpression of one of those AMPs have a negative effect on IIV-6, like decreasing the viral load per fly? I could imagine that the generation of da-GAl4 > UAS-Dipt or da-GAl4 > UAS-Drs flies, the infection of those flies with IIV-6 and performing RT-qPCR to measure the viral load could resolve this question. If the experiment should not for any reason be feasible, a thought experiment dealing with this point in the discussion section would be highly appreciated.

Whether the AMPs are directly antiviral is of interest to us as well, and this is a great idea. While we have not tested whether any of the AMPs are in fact antiviral, one study has shown that AttacinC is antiviral (Huang, Z.; Kingsolver, M. B.; Avadhanula, V.; Hardy, R. W., An antiviral role for antimicrobial peptides during the arthropod response to alphavirus replication. J Virol 2013, 87 (8), 4272-80.). Moreover, even if the individual AMPs are not antiviral in these over expression assays, this doesn’t exclude the possibility that other, currently uncharacterized, Relish targets are potent antivirals. Just as not every ISG is antiviral, it is possible that canonical Imd signaling—or non-canonical Relish activation via STING as has been suggested by others (see citations #13 & #14)—activates a transcriptional program with a variety of antimicrobial and antiviral targets. 

Compared to the in vitro data, the analysis using in vivo model of a previous nanostring data set (Fig. 3) suggest a shift in the timing of suppression. This issue should be shortly described in the discussion section.

We are unsure whether this is a genuine shift in suppression, as we do not know when the suppression begins in vivo. It is possible that we simply did not look early enough in vivo. A more thorough time course probing suppression in vitro and in vivo is needed to properly address this question. Moreover, we hypothesize that if there is a shift in the timing of suppression, it may be due to the fact that the in vitro infections occur at an MOI of 2, with 86~% of cells infected with rapid kinetics. In vivo, it may be that virus takes longer to diffuse through the hemolymph and infect permissive cells, and thus altering the kinetics of the response. 

I have some difficulties to discern the outcome of the nuclear localization of Relish. As the authors mentioned, the analysis faced several technical challenges. It seems that the number of IIV-6 infected cells is generally lower than in the uninfected state. The infection might change the properties of the cells, rendering them more sensitive to the preparation procedure. Probably, the YFP-Relish signal is clearer in the original images, but in the online and printed version, I have really difficulties to see the localization of YFP, except in the uninfected state where it is located in the cytoplasm. Thus, I would suggest adding panels just displaying the YFP, maybe as an inset in the existing images.

We agree that the YFP channel is difficult to see, and have created a new figure displaying the YFP channel on its own to the right of the corresponding overlay image.

It is generally well written, although one text part in the material and methods section seem to be copy-pasted and should therefore be changed for stylistic reason (L320-L322 and L326-L328).

Thank you for pointing this out. The Methods have been updated to be less redundant.

 Specific comments:

      line 27: “…perpetual arms race has created a plethora of mechanisms…” - I would suggest to introduce the term Red Queen Hypothesis from Van Valen (1973, A new evolutionary law. Evolutionary Theory 1: 1–30. ) in this context.

This citation has been added into the text where appropriate.

2.       line 34: the information that IIV-6 can infect Drosophila melanogaster should be added, also the reference (Bronkhorst et al., PNAS, 2012: The DNA virus Invertebrate iridescent virus 6 is a target of the Drosophila RNAi machinery).

This citation has been added

3.       line 46: Please remove “In order to begin” and start the sentence directly with “To address…”. The whole sentence is a bit bulky; please consider rephrasing it.

We have removed this superfluous transitionary phrase.

4.       line 51:  I am not familiar with the term cellular catastrophe. Could you please explain this term or use another word?

This comment has been removed.

5.       line 81 – line 82: Is the treatment with EcD also required for the expression of the Toll receptor? If this is the case, please add a short comment. If not, please explain why you add the treatment.

Yes, the Toll receptor is up-regulated with EcD treatment. We have updated the text to reflect this fact and have included a citation to our 2013 publication that includes this data from a microarray analysis. We can share this data, from our earlier publication, with the reviewers if needed.  

6.       line 85: The codeset of 139 immune-related genes should be included as list in the supplementary materials.

We have included this data as Supplementary Tables along with the raw expression count data from our two sets of Nanostring analyses. Additionally, the version of the codeset we used had only 102 genes, we have updated the text with this correction.

7.       line 102: Please add in Fig. 2 a short description of the color code (-3.00 to 3.00 as level of expression). Do this also for Fig. 3.

The heat map color codes indicate the Z score. The legends have been updated.

8.       line 104: Add the abbreviation for EcD+PGN and EcD+spz.

We have added these abbreviations.

9.       line 106: Remove “To begin”.

We have removed this superfluous transitionary phrase.

10.   line 108: Full stop/ period is missing after AMP genes [19].

Thank you for pointing out this grammatical error, a period has been added.

11.   line 125: Replace “must” with “is likely to”.

We have made this correction.

12.   line 136: Fig. 4, why do you change the experimental design and stimulate the samples just for 15 minutes? In the experiments before, the stimulation was 6 hours. Please explain.

While transcriptional responses take time for transcription to occur and for message to accumulate to detectable levels, signaling events occur immediately upon ligand detection. For this reason, we harvest RNA 6 hours after ligand stimulation to monitor immune gene expression, but protein is harvested 15 minutes after ligand stimulation to examine the immediate signal transduction events. 

13.   line 148: Please remove the brackets and rephrase the sentence in the bracket to fit in the text.

We have removed the brackets and rephrased.

14.   line 162: Replace “argue” with “suggest”.

We have made this correction. 

15.   line 214: Full stop/ period is missing after “survival curve [19]”.

Thank you for pointing out this grammatical error, a period has been added.

16.   line 309: The outcome of infection survivals depend also on the gender utilized. Please specify the gender you have used, and if you used males and females in the same survival, the ratio between the gender.

We used an equal mix of male and female flies in our experiments. The methods have been updated to state this.

17.   line 314 - 315: At which incubation temperature was ECC15 grown? Did you use a shaker, and if so, which rpm did you use? At which speed did you spun the culture?

Ecc15 was grown at 37C while shaking at 250RPMs. Cultures were spun in a microfuge at 13000g to pellet the bacteria. We have updated the methods to include these oversights.

18.   line 329: Please specify “standard lysis buffer”.

Our lysis buffer consists of 1% Triton X-100, 20mM Tris pH 7.5, 150mM NaCl, 2mM EDTA, and 10% glycerol, with 1mM DTT, 1mM Sodium Vandate, and 20 mM β-glycerolphosphate, and protease inhibitors. We have updated the methods section to include these details.

19.   line 337: which Hoechst did you use?

Thank you for pointing out this oversight. It is Hoechst 33342, and the text, methods, figures, and legends have been updated to reflect this.